# A De Novo Frameshift Mutation in RPL5 with Classical Phenotype Abnormalities and Worsening Anemia Diagnosed in a Young Adult—A Case Report and Review of the Literature

**DOI:** 10.3390/medicina59111953

**Published:** 2023-11-05

**Authors:** Moritz Dorenkamp, Naomi Porret, Miriam Diepold, Alicia Rovó

**Affiliations:** 1Department of Hematology and Central Hematology Laboratory, Bern University Hospital (INSELSPITAL), 3010 Bern, Switzerland; 2Division of Pediatric Hematology & Oncology, Department of Pediatrics, Bern University Hospital (INSELSPITAL), 3010 Bern, Switzerland

**Keywords:** bone marrow failure, pure red cell aplasia, erythroaplasia, Diamond–Blackfan anemia, RPL5, ribosomopathy, eltrombopag

## Abstract

Diamond–Blackfan anemia (DBA) is a congenital bone marrow failure syndrome associated with malformations. DBA is related to defective ribosome biogenesis, which impairs erythropoiesis, causing hyporegenerative macrocytic anemia. The disease has an autosomal dominant inheritance and is commonly diagnosed in the first year of life, requiring continuous treatment. We present the case of a young woman who, at the age of 21, developed severe symptomatic anemia. Although, due to malformations, a congenital syndrome had been suspected since birth, a confirmation diagnosis was not made until the patient was referred to our center for an evaluation of her anemia. In her neonatal medical history, she presented with anemia that required red blood cell transfusions, but afterwards remained with a stable, mild, asymptomatic anemia throughout her childhood and adolescence. Her family history was otherwise unremarkable. To explain the symptomatic anemia, vitamin deficiencies, autoimmune diseases, bleeding causes, and myeloid and lymphoid neoplasms were investigated and ruled out. A molecular investigation showed the RPL5 gene variant c.392dup, p.(Asn131Lysfs*6), confirming the diagnosis of DBA. All family members have normal blood values and none harbored the mutation. Here, we will discuss the unusual evolution of this case and revisit the literature.

## 1. Introduction

Diamond–Blackfan anemia (DBA) is a rare constitutional congenital bone marrow failure syndrome mostly caused by mutations in ribosomal genes resulting in ribosomopathy, which leads to progressive macrocytic normochromic anemia and erythroblastopenia [1]. Typically, the diagnosis is made at the median age of 2 to 3 months; 95% of DBA patients will be diagnosed before the age of 2 and 99% before the age of 5 [2]. The estimated incidence of DBA is between 1:100,000 and 1:200,000 live births without differences between ethnicities [3]. Approximately 45% of patients have inherited the mutation from a parent (usually autosomal dominant inheritance) and 55% have de novo mutations [1]. In the familiar form of DBA, there is a 50% chance that the pathogenic mutation is passed onto the offspring. Autosomal dominant DBA is caused by mutations in ribosomal genes (RPL/RPS, ribosomal protein, large/small). The clinical phenotype of DBA is rarely associated with the X-linked mutations of *GATA1* or *TSR2*, and there is only one case described with a recessive erythropoietin (EPO) mutation that presented with the typical histological findings of DBA. These entities are sometimes referred to as DBA-like [4]. A few DBA patients will present with congenital abnormalities. The most common malformations are craniofacial features (microcephaly, hypertelorism, and cleft palate), growth retardation, upper limb anomalies (classical triphalangeal thumb), genitourinary malformations, and heart defects. To characterize DBA patients, the diagnostic criteria have been defined [5]. The pathophysiology of DBA is not completely understood. RP deficiency, p53 activation, and additional mechanisms cause bone marrow failure and anemia (Figure 1). These mutations have a higher probability to lead to a specific phenotype, but this is not true for most findings [6]. Additionally, the penetrance of DBA is incomplete [7]. It is remarkable that the predominantly ribosomal mutations lead to a specific effect in hematopoiesis whilst causing relatively minor changes in the rest of the tissues. This can be explained by the high demand for ribosomal proteins due to the high proliferation of proerythroblasts and the accompanying globin synthesis. A reduction in the ribosome levels with normal ribosome composition has been shown to influence lineage commitment in hematopoiesis [8].

Another influencing factor is that mutations in some RP genes lead to a reduced expression of *GATA1*, which is an important erythroid transcription factor regulating processes in erythroid differentiation and maturation [9]. Furthermore, a reduction or a function impairment of *HSP70* proteins, which act as chaperones for *GATA1*, also contributes to ineffective erythroid differentiation. The tumor suppressor p53, through its relationship with *GATA1*, also plays a role in the pathophysiology of DBA. The models of human primary cells have shown that mutations in *RPS11* and *RPS19* are associated with the stabilization and increased phosphorylation of p53, leading to the activation of its targets, and ultimately promoting cell cycle arrest and apoptosis [10]. It is known that *GATA1* acts as an inhibitor of p53 [11]; its impairment constitutes another modality that can interfere with the normal development of hematopoiesis. Studies in cell cultures of cord blood CD34+ cells of DBA individuals (with mutations in *RPL5*, *RPL11*, and *RPS19*) could show a downregulation of its chaperone *HSP70*, leading to the increased phosphorylation of p53. The transduction of *HSP70* into *RPL11*-haploinsufficient CD34+ cells leads to a significant increase in the proliferation of burst forming units and colony forming units [12].

The reduced globin synthesis is another mechanism considered in the physiopathology of DBA and consequently, the relation of globin to heme diminishes, raising the amount of free intracellular heme [13]. Free intracellular heme has been proven to be toxic for cells through leading to higher levels of reactive oxygen species. The relevance of this mechanism is emphasized by the elevated levels of heme transporter FLVCR1 (feline leukemia virus subgroup C receptor-related protein 1), a membrane hem exporter in erythroid cells of DBA individuals [14].

The DBA phenotype and genotype is subject to ongoing research [15]. We present this case to highlight the existence of DBA with unusual worsening anemia in a young adult.

## 2. Case Presentation

A 21-year-old female patient of Tamil descent was presented in July 2021 to her family doctor due to orthostatic complaints. Persistent moderate macrocytic anemia was diagnosed. The hemoglobin was initially 82 g/L and the MCV 99.2 fl, and other values of the complete blood counts (CBC) were normal. A vitamin B12 deficiency was diagnosed and treated accordingly. After the substitution of cumulative 5 mg vitamin B12 over two months, the hemoglobin was 75 g/L with unchanged red blood cell (RBC) indices and the patient was referred to the hematology department of a tertiary hospital for further investigations. In June 2022, we saw the patient for the first time in the outpatient clinic; she was 21 years old and was in good health, but she complained of progressive fatigue. In the clinical examination, she was 145 cm tall and her weight was 40 kg (BMI 19.0). She has a series of congenital malformations (Figure 2) and no organomegaly or enlarged lymph nodes. The laboratory tests confirmed the macrocytic anemia with a hemoglobin of 74 g/L. The anemia was non-regenerative with an absolute reticulocyte count of 18 G/L, and the other CBC values were in the normal range. The ferritin, transferrin saturation, folic acid, and vitamin B12 were normal, but the holotranscobalamin (HoloTc) was reduced (37.4 pmol/L). In the evaluation of the peripheral blood film, macrocytosis was observed without other signs related to vitamin B12 deficiency. There was no evidence of hemolysis and the EPO was increased with a level of >750 mU/mL, which was interpreted in the context of anemia.

The review of her medical files showed that at birth, after an uncomplicated pregnancy, she had a ventricular septal defect, a double thumb on the right side, relative macrocephaly, and a lateral cleft lip. At the age of 10 weeks, she was hospitalized due to anemia-related heart failure. Her condition stabilized after transfusions. An elevated level of erythroid adenosine deaminase (eADA) activity suggested an underlying diagnosis of DBA, but at that time, genetic testing was not performed. As her hemoglobin levels were subsequently stable (above 80 g/L), therapies such as steroids were not administered. In childhood and adolescence, the hemoglobin values were always stable in the range of 85–100 g/L. There was a spontaneous closure of the ventricular septal defect around the age of 5 years with asymptomatic cardiac performance.

Due to her short stature (her height at birth was 44 cm, and weight was 2620 g, both <10th percentile, and both falling below the third percentile in childhood), she was subjected to somatropin supplementation from the age of 5 7/12 to 11 7/12. Under this regimen, she grew 28 cm (from 98 cm to 126 cm), after stopping this treatment, she reached a height of 145 cm (her father was 165 cm, and her mother was 155 cm). The cleft lip was surgically corrected with good results; the polysyndactyly does not limit the patient in her daily life.

To better characterize the worsening of the anemia observed since 2021, bone marrow (BM) investigations were performed. The aspiration showed scarce spicules and provided little information. The BM biopsy revealed normal cellularity for her age, but a highly atypical CD4-predominant T-lymphocytosis (60% of the total cellularity), an impairment of the maturation of all three cell lines, and mild myelofibrosis (maximum MF-1). The molecular pathological examinations showed no evidence of a clonal rearrangement of TCR gamma. A PET/CT scan showed no evidence of lymphoma. The karyotype analysis was unsuccessful due to insufficient metaphases, and recurrent findings typical of a myelodysplastic syndrome were not detected in fluorescence in situ hybridization (FISH). In the array-CGH, a copy neutral loss of heterozygosity of the short arm of Chromosome 3 and 8 was shown. Due to the suspicion of evolution in a myeloid disease, an NGS using the fragment analysis of 65 genes and 300 known recurrent rearrangements was evaluated. No somatic mutation was identified.

Finally, due to the consistent suspicion of a bone marrow failure syndrome, a heterozygous mutation in the RPL5 gene, variant c.392dup, p.(Asn131Lysfs*6), was detected with the TWIST Comprehensive Exome (Twist Bioscience) NGS panel. Combined with the clinical features of the patient, the diagnosis of DBA/Aase-Smith Syndrome II could be confirmed. In the family history, all relatives have a normal height according to their ethnic background; there was no history of malformations or anemia in her parents, sister, grandparents, uncles or aunts. The genetic testing of the parents and sister did not show the mutation and their CBCs were within the normal range (Figure 3).

During follow-up, a worsening of the anemia was observed. At that time, the diagnosis of sarcoidosis was suspected and she received short-term steroids (prednisolone). Once the diagnosis of sarcoidosis was ruled out, the steroid treatment was quickly discontinued. This short-term administration of the steroid did not improve the anemia. The nadir values of the hemoglobin were 58 g/L and RBC transfusions were needed. The transfusion requirement was of 2 RBC units every 5 months. A 3-month trial of Eltrombopag using an aplastic anemia dose was initiated and was well-tolerated but failed to improve the anemia and was therefore withdrawn. In the last 12 months, a further worsening of the anemia (nadir Hb 40 g/L) with a greater transfusion requirement of 2 RBC units every 3 months was observed. A new attempt with steroids (0.5 mg/d) was made, and this time for longer time. A total of 28 days after the start of therapy, a reticulocyte response of 51 G/L (previously 10 G/L) with the stabilization of Hb values above 80 g/L was observed. Currently, steroid dose tapering is ongoing. The relevant aspects of this case are summarized in Table 1.

## 3. Discussion

We present here a DBA case in which anemia was diagnosed a few weeks after birth, but afterwards, the patient had an indolent course until the age of 20, when the patient began to show a clear worsening of the anemia, thus requiring transfusions and treatment. Although, due to her malformations, a congenital syndrome was suspected since birth, a confirmation diagnosis was not made until referral to our center for an evaluation of the anemia. After the exclusion of a myelodysplastic syndrome, lymphoma, and inflammatory diseases, a heterozygous gene variant in the *RPL5* gene, c.392dup, p.(Asn131Lysfs*6), was detected. This frameshift mutation leads to an early stop codon, and thus, nonsense-mediated mRNA decay. Pathogenic variants in the *RPL5* gene are associated with DBA [7]. The clinical, laboratory, and molecular investigation of the family did not show the mutation found in the proband in any of the other members, allowing a de novo diagnosis, which is described in 55% of DBA patients [1].

The majority of DBA cases will present with severe anemia during the first year of life and they will require continuous treatment [15]. There are also many patients who are asymptomatic carriers, although cases of late-onset anemia during adulthood have been frequently reported [16,17,18]. Cases such as the one reported here, diagnosed in the first weeks of life and following a mild course, to then worsen again in adulthood, seem to be rare.

Furthermore, patients with inherited bone marrow failure syndromes (IBMFSs) are more likely to develop secondary malignant diseases compared to the general population. In DBA, this is not only true for MDS and AML, with an observed/expected ratio of 28.8 and 352.1, respectively [19], but also for solid tumors such as colon carcinoma, esophageal cancer, osteogenic sarcoma, squamous cell carcinoma and lung cancer. The O/R ratio for all cancers is reported to be 4.75. In other IBMFSs, like Fanconi anemia and dyskeratosis congenita, higher O/R ratios are reported but they lack the high incidence of colon cancer and osteogenic sarcoma, implying a different cancer mechanism associated with RP mutations [19]. Based on this evidence, the first hypothesis in the case presented was the progression to a myeloid disease or the appearance of other neoplasms, for which various investigations were carried out. The patient did not show either typical morphological signs or typical mutations of MDS. The PET scan ruled out an active malignant process. A gynecological examination and ultrasound were both normal.

Regarding treatment for DBA, 80% of patients initially respond to steroids. A trial of 2 mg/kg/d is frequently used; reticulocytes will typically appear after two weeks in responders. If there is no response after four weeks, the treatment should be discontinued. If there is an increase in reticulocytes and hemoglobin levels, steroids should be decreased to the minimal dosage to maintain a hemoglobin of 90 g/L. This dose should not exceed 0.3 mg/kg/d if erythrocyte transfusions are easily available (0.5 mg/kg/d if they are not) [1]. Due to the long-term effects of steroids and transfusions, the option to perform an allogeneic hematopoietic stem cell transplantation (allo-HCT) should be evaluated early. In 2020, Strahm et al. evaluated the data of 70 DBA patients that underwent allo-HCT in France and Germany from 1985 to 2017 [20]. They concluded that allo-HCT is efficient and safe in young patients (median age of 5.5 years) with steadily improving outcomes, e.g., no case of death or severe chronic GVHD after 1999 in patients who were transplanted from a matched sibling donor. The probability of survival with a median follow-up of 4.5 years was 91% with a low transplant related mortality rate. Six of 70 patients died of transplantation-related causes. The surviving patients were independent of tranfusions [20]. In this study, the difference in outcome for patients >10 years at transplantation did not reach statistical significance, while an Italian study of 30 patients (1990–2012) showed significantly better outcomes for patients younger than 10 years old with all transplantation-related deaths occurring in the older patients [21], who had severe iron overload before transplantation. Thus, allo-HCT should be considered early in young patients, preferably if a matched sibling donor is available. The patient has a HLA-identical sister, but so far, there is no indication for an allo-HCT.

Eltrombopag is an orally available mimetic of thrombopoetin, which is a regulator of platelet production and the maintenance of human stem cells. Currently, eltrombopag is used for immune thrombocytopenic purpura [22], thrombocytopenia due to hepatitis C [23], and for the treatment of severe aplastic anemia [24,25]. The hematopoietic effects are achieved through its interaction with the TPO receptor (c-mpl) and the activation of signaling cascades in its target cells [26]. Additionally, the drug is known for its capacity to chelate intracellular iron and thus reduce the labile iron pool, both decreasing reactive oxygen species and inhibiting heme synthesis. This makes it an interesting drug to address the heme/globin disbalance found in DBA progenitor cells. In ex vivo models of induced pluripotent stem cells of DBA patients, a partial rescue of the defective erythropoiesis using eltrombopag, with improved differentiation and maturation, could be shown [27]. The effect turned out to be even larger with the addition of deferasirox compared to deferoxamine, which is a less effective chelator of intracellular iron [28]. In 2020, a prospective phase 2 study of eltrombopag use in patients with moderate aplastic anemia or uni-lineage cytopenia investigated the erythropoetic response and clonal evolution under eltrombopag. One DBA patient with an *RPS19* mutation who was refractory to steroids and transfusion-dependent was included and became transfusion-independent even 10 months after the discontinuation of eltrombopag before relapsing. This patient regained a response after re-establishing the treatment for the whole follow up of 5 years [29]. These findings have led to the approval of an interventional single-arm phase I/II trial that will assess the safety and efficacy of eltrombopag in DBA patients, which will be completed in 2027 (NCT04269889).

## 4. Conclusions

More awareness of unusual DBA presentations is needed. While testing for the causative mutation in DBA is not necessary to establish the diagnosis, it is important for the follow-up of these patients, due to the increased risk of transformation into myeloid diseases and other tumors, as well as for genetic counseling. Multidisciplinary clinical follow-up is also necessary for the management of non-hematological problems. Furthermore, more data about the underlying genetic profile will allow for a better understanding of the geno- and phenotype relationship in this heterogenic disease.

## Figures and Tables

**Figure 1 medicina-59-01953-f001:**
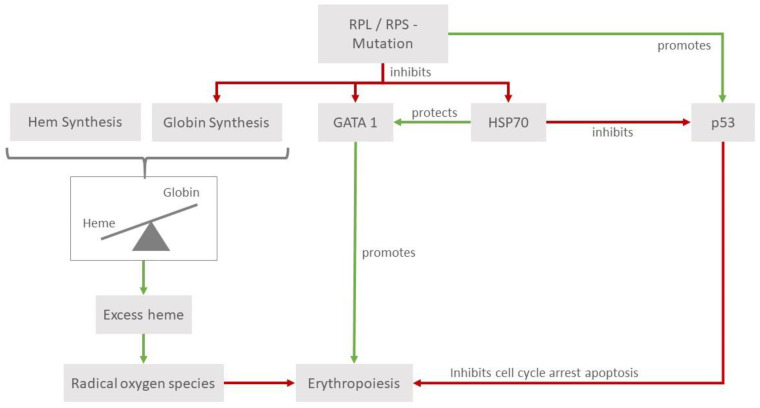
Pathophysiology of *RPL* and *RPS* mutations. *RPL* and *RPS* mutations can affect erythropoiesis in various ways that are emphasized in hematopoietic tissues. They lead to higher levels of p53, which, in its role as a tumor suppressor, leads to cell cycle arrest, thus impeding erythropoiesis. Reduced levels of heat shock protein 70 (*HSP70*) affect erythropoiesis through its interactions with the erythroid transcription factor *GATA1* and p53. Additionally, the impaired balance in the heme/globin ratio leads to a detrimental increase in radical oxygen species.

**Figure 2 medicina-59-01953-f002:**
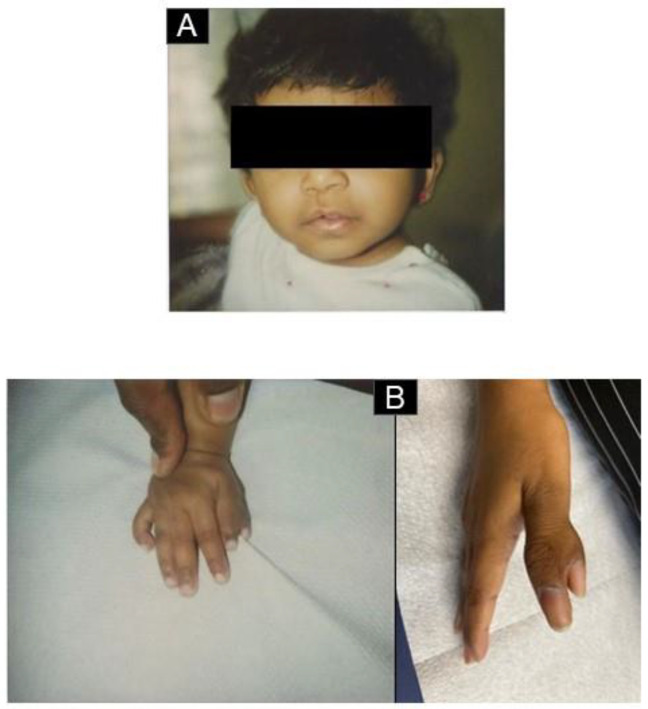
Congenital anomalies. (**A**) Lateral cleft lip after surgical correction. (**B**) Polysyndactylie (“double thumb”), 2 pictures: childhood/adulthood. (**C**) Growth curve of the patient. As she was not treated with steroids in infancy, the accentuated growth retardation was partially corrected with the administration of somatotropin between the ages of 6 and 10 years’ treatment [15].

**Figure 3 medicina-59-01953-f003:**
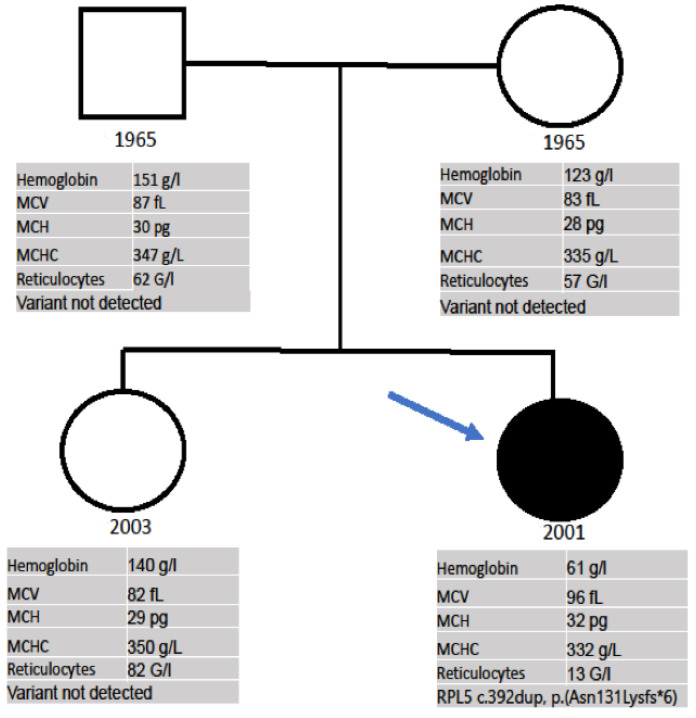
Pedigree of the family. Proband is indicated by arrow. Symbols: □ = male, ○ = female; • = heterozygous mutation in the RPL5 gene, variant c.392dup, p.(Asn131Lysfs*6).

**Table 1 medicina-59-01953-t001:** Summary of patient characteristics.

Feature	Patient Female, Born 2001
Clinical onset of anemia	At ten weeks of age
Heart anomalies	Ventricular septal defect, spontaneous closure at 5 years of age
Facial anomalies	Hypertelorism, relative macrocephaly, lateral cleft lip
Growth anomalies	145 cm after prepubescent growth hormone treatment without corticosteroids
Limb anomalies	Polysyndactyly of the thumb
Adenosine deaminase	Elevated
Steroid responsive	Yes (first trial at 21, second trial at 22 years)
Bone marrow histology	(Atypical for DBA)—normal cellularity according to age but CD4-lymphocytosis (60% of cellularity), dysplastic megakaryopoiesis, reduced erythropoiesis and myeloid line
Genetics	*RPL5* c.392dup, p.(Asn131Lysfs*6) No somatic myeloid mutations *

* NGS and fragment length analysis examined 65 genes using the TruSight Oncology500 panel from Illumina, San Diego, CA 92122, USA. Using Illumina’s TSO500 gene panel, 1.28 Mb of genomic sequence was analyzed. Sequencing was performed with the Novaseq 6000 instrument (Illumina). Of these, 65 relevant myeloid genes were evaluated. Detection limit 5%, variant allele frequency (VAF). Human reference genome: GRCh37/hg19. Full-length mutations in the following genes: ASXL1, ASXL2, ATRX, BCOR, BCORL1, BRAF, CALR, CBL, CDKN2A, CEBPA, CREBBP, CSF3R, CSNK1A1, CTCF, CTNNA1, CUX1, DDX41, DNMT3A, EP300, ETV6, EZH2, FBXW7, FLT3, GATA1, GATA2, GNAS, HRAS, IDH1, IDH2, IKZF1, JAK2, KDM5A, KDM6A, KIT, KMT2D, KMT2C, KRAS, MLL, MPL, MYC, MYD88, NF1, NPM1, NRAS, PHF6, PPM1D, PTEN, PTPN11, RAD21, RB1, RUNX1, SETBP1, SF3B1, SH2B3, SMC1A, SMC3, SRSF2, STAG2, SUZ12, TET2, TP53, U2AF1, WT1, ZBTB7A, ZRSR2. Rearrangements with following genes were evaluated: ABL1, ALK, BCL2, BRAF, CCND1, CREBBP, ETV6, EGFR, FGFR1, FGFR2, FUS, HMGA2, JAK2, KMT2A (MLL), MECOM, MET, MLLT3, MLLT10, MYBL1, MYH11, NTRK3, NUP214, PDGFRA, PDGFRB, RARA, RBM15, RUNX1, TCF3, TFE3. The rearrangements were analyzed using the Oncomine Myeloid Research Assay from ThermoFisher Inc., Waltham, MA, USA. Sequencing was performed using Ion Torrent S5. The panel includes over 300 known recurrent rearrangements, e.g., RUNX1-RUNX1T1, CBFB-MYH11, PML-RARA, MLLT3-KMT2A (and numerous other MLL or KMT2A Rearrangements), DEK-NUP214, RBM15-MKL1, and BCR-ABL1, as well as >300 rare rearrangements, with the following partners: ABL1, ALK, BCL2, BRAF, CCND1, CREBBP, ETV6, EGFR, FGFR1, FGFR2, FUS, HMGA2, JAK2, KMT2A (MLL), MECOM, MET, MLLT3, MLLT10, MYBL1, MYH11, NTRK3, NUP214, PDGFRA, PDGFRB, RARA, RBM15, RUNX1, TCF3, TFE3.

## Data Availability

All data generated for this manuscript is included in the published article.

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
