# Peer review of "A De Novo Frameshift Mutation in RPL5 with Classical Phenotype Abnormalities and Worsening Anemia Diagnosed in a Young Adult—A Case Report and Review of the Literature"

_medicina, 2023, doi:10.3390/medicina59111953_

Round 1

Reviewer 1 Report

Comments and Suggestions for Authors

This is a well written paper reporting on a rather rare condition such as DBA and a de novo frameshift mutation in RPL5 gene in a young woman diagnosed at the age of 21 yrs. The phenotype of the patient was typical and the haematological findings as well. It is noteworthy  that the patient remained  transfusion-independent for several years throughout childhood and adolescence and manifested severe anemia later in her life even though mild anemia was the constant finding of her medical history. 

Comments:

1) How many centimetres did she gain after somatotropin therapy? 

2) Did she exhibit other morphological abnormalities of the peripheral blood  due to B12 deficiency (hypersegmentation of neutrophils) ? Or only the measurement was low?

3) Why she was considered as non-responsive to corticosteroids at the first time? What was  the initiation dose and for how long? What about the second time she started again on corticosteroids with low dose (0.5mg/kg) and she responded by rising the Hb level? (Needs clarification)

4) It should be stressed out that DBA patients need multispecialty clinical observation and care as the haematological problem is not the only one.

Author Response

We thank the editor and reviewers for their valuable comments and the opportunity to revise our manuscript. Please find our point-to-point response below. Corresponding changes within the text are highlighted in yellow.

Reviewer 1.

This is a well written paper reporting on a rather rare condition such as DBA and a de novo frameshift mutation in RPL5 gene in a young woman diagnosed at the age of 21 yrs. The phenotype of the patient was typical and the haematological findings as well. It is noteworthy  that the patient remained  transfusion-independent for several years throughout childhood and adolescence and manifested severe anemia later in her life even though mild anemia was the constant finding of her medical history. 

Comments:

We would like to thank reviewer 1 for the important comments on this manuscript. We have worked on each of the following mentioned points, making proper changes to the text.

  • How many centimetres did she gain after somatotropin therapy? 

Thank you for this question. Under somatotropin the patient grew 28 cm (from 98 to 126 cm). After stopping this treatment, she grew further reaching a height of 145 cm.

We have added this information in the manuscript as follow:

“Under this regimen, she grew 28 cm (from 98 cm to 126 cm), after stopping this treatment,  she reached a height of 145 cm (father 165 cm, mother 155 cm).”

  • Did she exhibit other morphological abnormalities of the peripheral blood due to B12 deficiency (hypersegmentation of neutrophils) ? Or only the measurement was low?

Thank you for this question. She had macrocytosis but not hypersegmentation of neutrophils. We have added in the text:

“In the evaluation of the peripheral blood film, macrocytosis was observed without other signs related to Vitamin B12 deficiency.”

  • Why she was considered as non-responsive to corticosteroids at the first time? What was  the initiation dose and for how long? What about the second time she started again on corticosteroids with low dose (0.5mg/kg) and she responded by rising the Hb level? (Needs clarification)

Thank you very much for this suggestion. The first cycle of steroids was given due to the suspected diagnosis of sarcoidosis, which was quickly ruled out, and after a week, the corticosteroids were suspended and we observed that there was no improvement in the anemia. The second attempt was intended to treat anemia, and a 0.5 mg/d regimen was used for a longer period, in fact, she is still under this treatment (tapering) and showed a reticulocyte response and did not require any further RBC transfusions.

We have added in the manuscript the following paragraph:

“During follow-up, a worsening of the anemia was observed. At that time, the diagnosis of sarcoidosis was suspected and she received a short-term steroids (prednisolone). Once the diagnosis of sarcoidosis was ruled out, steroid treatment was quickly discontinued. This short-term administration of steroid did not improve the anemia.”  

Reviewer 2 Report

Comments and Suggestions for Authors

I classified as of high significancy, soundness, interest and merit because it is  a very nice case report and review calling the attention for rare diseases often overlooked in clinic. Every now and then it is important to address this subject and the authors have done it with a comprehensive approach. It needs, however, a thoroughly English review.

There are some misplaced words, sometimes making it difficult to understand what the authors want to say.

My only criticism other than a review in English is the tittle: as it is, it gives the idea that the disease presented itself in adulthood which is not the case. The diagnosis was not done in childhood because it was not considered at the time and hematological symptoms were mild and uneventful during childhood. This is my only criticism to the entire paper. 

My suggestion for the title is as follows:

A de novo Frameshift mutation in RPL4 with classical phenotype abnormalities and worsening anemia diagnosed in a young adult - a case report and review of the literature. 

Comments on the Quality of English Language

It is my strong recommendation already mentioned in the comments above.

Author Response

We thank the editor and reviewers for their valuable comments and the opportunity to revise our manuscript. Please find our point-to-point response below. Corresponding changes within the text are highlighted in yellow.

Reviewer 2.

I classified as of high significancy, soundness, interest and merit because it is  a very nice case report and review calling the attention for rare diseases often overlooked in clinic. Every now and then it is important to address this subject and the authors have done it with a comprehensive approach. It needs, however, a thoroughly English review.

Comments

We would like to thank reviewer 1 for the important comments on this manuscript. We have worked on each of the following mentioned points, making proper changes to the text.

There are some misplaced words, sometimes making it difficult to understand what the authors want to say.

We sincerely appreciate the reviewer’s comments. A native English speaker has carefully revised and edited the paper. 

My only criticism other than a review in English is the tittle: as it is, it gives the idea that the disease presented itself in adulthood which is not the case. The diagnosis was not done in childhood because it was not considered at the time and hematological symptoms were mild and uneventful during childhood. This is my only criticism to the entire paper. 

My suggestion for the title is as follows:

A de novo Frameshift mutation in RPL4 with classical phenotype abnormalities and worsening anemia diagnosed in a young adult - a case report and review of the literature. 

Thank you very much for this suggestion. The title has been accordingly adapted:

 “A de novo frameshift mutation in RPL5 with classical phenotype abnormalities and worsening anemia diagnosed in a young adult – a case report and review of the literature”
